# Psychoeducation Improved Illness Perception and Expressed Emotion of Family Caregivers of Patients with Schizophrenia

**DOI:** 10.3390/ijerph18147522

**Published:** 2021-07-15

**Authors:** Watari Budiono, Kevin Kantono, Franciscus Cahyo Kristianto, Christina Avanti, Fauna Herawati

**Affiliations:** 1Department of Pharmaceutics, Faculty of Pharmacy, University of Surabaya (Ubaya), Surabaya 60293, Indonesia; watari.b@gmail.com (W.B.); fckristianto@staff.ubaya.ac.id (F.C.K.); christina@staff.ubaya.ac.id (C.A.); fauna@staff.ubaya.ac.id (F.H.); 2Department of Food Science, Auckland University of Technology, Private Bag 92006, Auckland 1142, New Zealand

**Keywords:** family intervention, schizophrenia, psychoeducation, medication adherence, Indonesia, illness perception

## Abstract

Social interventions such as psychoeducation, in conjunction with appropriate antipsychotic medications, positively impact schizophrenic patients’ recovery. The aim of this 12-week study was to compare standard Indonesian mental healthcare for schizophrenia with psychoeducation-enriched care for family members, investigating both family and patient parameters. Sixty-four family participants meeting pre-set criteria were recruited from various online Indonesian community forums, social media, seminars/gathering events, and inpatient visits. Each family member was the main care provider for one patient with a schizophrenia diagnosis. Family participants were randomly allocated to one of two groups (control or intervention); both groups received equal personal time and attention from staff but the control group lacked the specific psychoeducational aspect of the intervention. In comparison with the control group, pre- and post-evaluation revealed significant positive effects in the intervention group for illness perception (F_(ave)_ = 124.85; d_(ave)_ = 2.72) and expressed emotion (OR_(ave)_ = 0.39) among family members. For the patients, there was a significant positive effect on medication adherence (F_(1, 62)_ = 21.54; *p* < 0.001, d_(intervention)_ = 1.31) if their family members were in the intervention group. Partial least-squares path modeling revealed that low expressed emotion in family members was positively correlated with high medication adherence (β = −0.718; *p* < 0.001) in patients. This study provides evidence for the patient and family benefits of family psychoeducation on schizophrenia in a diverse Indonesian population.

## 1. Introduction

Family members act as primary caregivers for patients with schizophrenia, offering care and support as well as ensuring medication adherence. It has been reported that family burden can change the quality of care that families provide [1] and that this is based, at least in part, on their perception of the nature and cause of the patient’s illness and their emotional reactions to the patient’s symptoms [2,3,4,5,6]. Family psychoeducation about the nature of schizophrenia illness has been shown to be effective in positively changing family members’ reactions and indirectly increasing patients’ adherence to their prescribed treatment [7,8]. Our aim was to confirm these findings for the first time in the Indonesian setting. We hypothesized that an add-on psychoeducational program for family members would aid family members’ illness perception and reduce their expressed emotion, and that this, in turn, would increase the patients’ medication adherence.

## 2. Materials and Methods

### 2.1. Ethics Statement

This study has been approved by the University of Surabaya (Ubaya) ethics committee. Data collection took place between June and December 2016. Participants were recruited from online Indonesian community forums, online social media, seminars/gathering events, and in-patient visits. Family members of patients with schizophrenia were invited to approach the researcher regarding a psychoeducation program to further understand schizophrenia. Information sheets outlining the study procedure were provided to the participants who expressed interest. All participants, both family members and patients, provided written consent. They were assured that they could drop out of the study at any time without prejudice to the care of their patient. They understood that, if they were in the control group, they could be provided with the psychoeducation intervention at the completion of the study. 

### 2.2. Study Design

A total of 64 family members were recruited for this study (Figure 1). Each was a primary caregiver of one patient participant. Original recruits who did not meet eligibility criteria were excluded but were provided with the psychoeducational support applied in the study. Eligibility criteria were: (1) one member of the family diagnosed with Diagnostic and Statistical Manual of Mental Disorders, 5th Edition (DSM-5) schizophrenia, (2) the family member caring for one patient, and (3) the patient having had symptoms of schizophrenia for at least six months. Participants who were eligible were then assigned a three-digit random code and randomized into the two groups (i.e., control and intervention). Trained therapists were randomly assigned to families. As per the consent form, participants in the control group (family member and patient) received the same amount of time and attention from their therapists as those in the intervention group but family participants were provided with no additional psychoeducational material (they received the psychoeducation at the completion of the study). The intervention consisted of educational video materials about schizophrenia. A total of six educational videos were provided every two weeks to each participating family member to watch at home. The video topics were: (1) introduction, (2) stigma in schizophrenia, (3) current schizophrenia therapies, (4) expressed emotion, (5) family and caregiver challenges, and (6) patients’ mood state. Each participant received a unique link to view the video and a video counter (only visible to the researchers) was added to confirm that participants viewed the video. Participants were given a phone/text/email reminder to watch the video periodically throughout the study.

The first module of psychoeducation was a general introduction to schizophrenia, which included the definition of schizophrenia, potential symptoms, types of schizophrenia, potential causes, and current therapies. The second module detailed challenges in managing schizophrenia, which covered common myths, stigmatization in Indonesian culture and society, and self-stigmatization, focusing on its consequences and how to manage it. The third module presented current therapies for patients with schizophrenia, discussing antipsychotic medication, its mechanism of action, side effects, and potential causes of patients’ lack of medication adherence. The fourth module provided an explanation of expressed emotion, highlighting its definition and three dimensions (criticism, hostility, emotional overinvolvement), along with its association with relapse. The module proposed more appropriate ways to communicate and manage conflict and symptoms, as well as supporting the audience’s understanding of relapse triggers. The fifth module addressed the challenges that the family may face in the context of Indonesian culture. This module explained the importance of family support and provided the results of research into potential causes of schizophrenia. In the final module, there was a recap of the previous material and more detailed information was then given on the influence of schizophrenia on patients’ ability to regulate their mood. 

### 2.3. Data Collection Instruments

The illness perception questionnaire for schizophrenia relatives (IPQS-R) [1] was administered to measure family members’ illness perception. The IPQS-R consists of 13 subscales measuring identity (*n* = 18 items), cause (*n* = 26 items), acute/chronic timeline (*n* = 6 items), cyclical timeline (*n* = 4 items), consequences for the patient (*n* = 11 items) and relative (*n* = 9 items), patient’s personal control (*n* = 4 items) and their relative’s (*n* = 4 items), personal blame by the patient (*n* = 3 items) and their relative (*n* = 3 items), treatment control/cure (*n* = 5 items), illness coherence (*n* = 5 items), and emotion representation (*n* = 9 items). 

Five-minute speech samples (FMSS) [9] were used to measure family members’ expressed emotion [2]. The patient’s medication adherence was measured using an adapted version of the Morisky Medication Adherence Scale (MMAS-4) [3], which was specifically developed for this study due to language and cultural barriers. The adapted scale was called the Indonesian Medication Adherence Scale (IMAS), similar to that used in [6]. The IMAS is a four-item questionnaire with a Yes/No scoring scheme; an IMAS score of 0–1 indicates lack of or no adherence, 2 reflects medium adherence, and 3–4 reflects high adherence.

All data collection instruments were originally developed in English and were translated to Indonesian. The translation was done by a local certified translator. The translation followed the WHO’s process of translation and adaptation of instruments, where a forward translation was first performed with a follow-up back-translation process afterward. To further validate the translation as well as improve the overall clarity of the survey questions, a pre-test was carried using a focus group of 10 participants.

### 2.4. Family Members’ FMSS Assessments

The family members of the patients were contacted in a single session using a voice interview through Skype or another virtual call service. Video interviews were not conducted as the majority of the participants were concerned about their privacy, as mental illnesses in Indonesia are largely stigmatized in society. A study done by Tompson et al. [4] has shown no significant differences in terms of expressed emotion ratings for family members interviewed on the telephone using FMSS. Recent studies have also utilized phone FMSS interviews to understand the treatments for maternal depression [5], schizophrenia [10], binge eating [11], and anorexia nervosa [12]. In this study, the FMSS lasted approximately 10 min [2]. FMSS for expressed emotion are scored on four categories: (1) quality of initial statement, (2) relationship rating, (3) number of critical comments, and (4) emotional overinvolvement. An individual will be rated as having high expressed emotion (critical) if there is: (1) a negative initial statement, (2) an overall negative rating for the patient and family member’s relationship, or (3) one or more critical comments made about the patient. An individual will be rated as having high emotional overinvolvement (EOI) if they: (1) report self-sacrificing/overprotective behavior, (2) give an emotional display during the interview, or (3) if there is a combination of two of the following: excessive detail about the past, one or more statements of positive attitude, or excessive praise (five or more positive remarks). Three different trained raters for FMSS rated participants’ expressed emotion for both criticism and EOI. 

### 2.5. Data Analysis

A 2 × 2 factorial design was used in this study, and so a repeated measures ANOVA (RM-ANOVA) was applied with independent variables of pre/post and control/intervention to examine the effectiveness of intervention for caregivers’ or family members’ illness perception and expressed emotion and patients’ medication adherence. The Bonferroni correction for multiple tests was applied. Checks such as data normality and homogeneity of variance and covariances were carried out to ensure normal assumptions. Effect size estimates were calculated using standardized mean pre-test and post-test differences in the control and intervention groups.

In addition, mean inter-item and inter-subscale correlation (i.e., Pearson correlation), Cronbach Alpha, and Kaiser-Meyer-Olkin (KMO) tests were carried out to assess the validity, reliability, and sampling adequacy in this study. K statistics were applied to the FMSS expressed emotion ratings to assess their reliability. Odds ratios were also calculated for the expressed emotion ratings. 

In addition to a classical univariate approach, this study utilized partial least squares path modeling (PLS-PM) to construct cause-and-effect models. PLS-PM summarized the relationships between family members’ illness perception and their expressed emotion, and deduced how that then influenced patients’ medication adherence. PLS-PM was suitable in this study as it can generate both normal and non-normal latent models, is suitable for studies with small to medium samples, and is relatively robust to multicollinearity concerns. In this study, both the control and intervention pre-post group data were modeled, with a total of 128 observations included. Goodness of Fit (GoF) statistics, Cronbach’s α, and Dillon-Goldstein’s rho were obtained for each manifest variable to assess the model reliability. Additionally, the R^2^ value of the latent variable was also calculated. A value greater than R^2^ = 0.7 for all three statistics indicated sufficient reliability. All analysis was carried out using SPSS v. 20 (IBM Corporation, New York, NY, USA) and XLSTAT 2021.2.2 (Addinsoft Inc, New York, NY, USA).

## 3. Results

### 3.1. Patient Characteristics

All 64 patients had a diagnosis of DSM-V schizophrenia, made clinically by an experienced clinician. The patients were mostly male (*n* = 43), with a mean age of 34.03 years (SD = 7.05 years). The average length of contact with mental health services was 6.19 years (SD = 0.96 years). Approximately 50% (*n* = 32) were receiving clozapine, 31% (*n* = 20) were receiving phenothiazines, and the rest were receiving aripriprazole. In all cases, psychotic symptoms had been stable for at least six weeks. 

### 3.2. Family Members’ Characteristics

Family members who participated in this study were balanced in terms of gender (male = 38; female = 26). Their mean age was 43.61 years (SD = 14.36 years), with most being the parents of the patient (*n* = 34); the remainder were spouses (*n* = 15), siblings (*n* = 12), or the patient’s children (*n* = 3). Of the family members, 43 lived in the same house as the patient. The number of waking hours they reported spending with their schizophrenic relation each week ranged from 20 to 100 ($\bar{x}$ = 59.73, SD = 22.83 h). 

### 3.3. IPQS-R 

The mean scores on the identity subscale suggested that most participants identified with more than half of the experiences listed (Table 1). All 18 identity items were equally selected, with more than 50% of the items experienced by participants. The most frequent items were paranoia (72%, *n* = 70), irritation, worry, nervousness (equal scores for all three: 67%, *n* = 65), having thoughts they would rather not have (64%, *n* = 62), and being suspicious of other people (64%, *n* = 62). The least frequently selected items were loss of motivation (53%, *n* = 51), hearing voices (57%, *n* = 55), and not doing much (53%, *n* = 58). Most of the experiences were attributed to the category of mental health problems. Overall, family members viewed the patient’s mental health problems as being chronic and cyclical in nature ($\bar{x}$ = 3.69) and perceived there to be a high degree of negative consequences for both the patient and their family members resulting from the patient’s mental health problems ($\bar{x}$ = 3.71). Little sense of control over the problems affecting both the patient and family members ($\bar{x}$ = 1.78) was reported with high personal blame by the patient ($\bar{x}$ = 3.91) and less by their family members ($\bar{x}$ = 2.02), along with slightly lower reported belief in the effectiveness of the treatment by family members ($\bar{x}$ = 2.48). Patients generally reported medication adherence ($\bar{x}$ = 3.25). There was a negative overall emotional response to having mental health problems ($\bar{x}$ = 3.67). Ideas on the causes of mental health problems ($\bar{x}$ = 4.2) were stress or worry, trauma, a chemical imbalance, the person’s mental attitude, or thinking about things too much. The beliefs least likely to be endorsed ($\bar{x}$ = 2.3) were that mental health problems are due to pollution in the environment, a germ or virus, taking illicit drugs, or having a drink spiked with illicit drugs.

Cronbach’s alpha was calculated for the subscales that were designed to assess coherent dimensions. Following the Streiner [13] criteria, all of the alphas were in the desired range ($\bar{x}$_α_ = 0.85; 0.7–0.9). Further analysis of the KMO and inter-subscale correlations can be found in Appendix A.

### 3.4. Family Members’ FMSS Expressed Emotion Ratings

In general, the expressed emotion in both the control and intervention groups was classified as high prior to intervention, and family members were categorized as having high levels of criticism and emotional overinvolvement. Results from chi-square analysis showed that there were no significant changes in terms of participants’ expressed emotion when it came to criticism or EOI in the control group (Table 2). Yet, more than half of the participants that were classified as having high expressed emotion for criticism and EOI in the intervention group demonstrated lower expressed emotion post-intervention in terms of both factors (criticism OR: 0.32; EOI OR: 0.46). 

### 3.5. Medication Adherence Scale

Participants were also asked to complete the IMAS to further investigate the patients’ medication adherence. Overall, the medication adherence in both the control and intervention groups was low prior to intervention ($\bar{x}$_control_ = 0.5 ± 1.3; $\bar{x}$_intervention_ = 1.41 ± 1.2). All participants in the control group rated as more than two on the IMAS. However, only half of the participants in the intervention group rated as more than two. The overall model showed a significant difference (*F*_(1, 62)_ = 21.54; *p* < 0.001). In contrast to each other, the control group did not show any difference ($\bar{x}$_pre_ = 0.5 ± 1.3, $\bar{x}$_post_ = 0.78 ± 1.1; d = 0.23), while the intervention group showed a significant increase in medication adherence ($\bar{x}$_pre_ = 1.41 ± 1.2, $\bar{x}$_post_ = 2.62 ± 0.5, d = 1.31). This clearly showed that psychoeducation had a positive impact on patients’ adherence, with a significant increase in medication adherence post-intervention. 

### 3.6. Overall Relationship between Illness Perception, Expressed Emotion, and Medication Adherence

Based on the PLS-PM model (Figure 2), it was shown that caregivers’ expressed emotion has a higher negative effect (β = −0.718; *p* < 0.001) on medication adherence than caregivers’ illness perception (β = 0.149; *p* < 0.003), although both were found to be significant contributors to medication adherence. In addition, illness perception was found to be significant for a person’s expressed emotion (β = 0.307; *p* < 0.001). Our results highlight that improvement of both the illness perception and expressed emotion of caregivers is crucial to improving patients’ medication adherence.

## 4. Discussion

### 4.1. Family Members/Caregivers’ Illness Perception

Our psychoeducational intervention significantly lowered the family’s negative perception of schizophrenia and significantly increased their awareness of illness (Section 3.3). We attribute this to the intervention. This is important in Indonesia because most of the population stigmatizes illnesses such as schizophrenia. One common belief here is that mental illness is caused by possession by an evil spirit; this results in the practice of *pasung* [14]—physical restraint and confinement of the patient—even though this has been outlawed since 1977. Participants in this study were provided with an opportunity for informal discussion with the researchers. They provided positive feedback about the intervention, finding it very useful. This perhaps highlights the lack of knowledge in this subject area held by caregivers in Indonesia. Our results agree with those of other studies [15,16] that have investigated the relationship between illness perception and quality of life related to expressed emotion (EE) in relatives of Mexican patients with psychosis, as well as relating EE with patients’ medication adherence [17]. 

### 4.2. Expressed Emotion

In this study, family members’ expressed emotion was shown to significantly decrease below the baseline after psychoeducation in the intervention group but not in the control group without education (Section 3.4). This was one of the main goals of the intervention. Numerous studies have shown that the quality of the relationship between the caregiver and patient is one of the key determinants of the outcome. Insufficient information is thought to be an important cause of familial expressed emotion [7,18,19]. 

### 4.3. Medication Adherence

Patients’ medication adherence significantly increased (Section 3.5) in the intervention group but not in the control group. This can be attributed to family members’ increased knowledge of the illness, although many studies have not been able to confirm this effect [8].

### 4.4. Psychoeducation in Indonesia

Our study demonstrates the effectiveness of psychoeducation and highlights the need to custom-tailor this to families and patients with schizophrenia in Indonesian communities. This is in agreement with the study by Chien [20], which highlighted the lack of family intervention studies in Asian populations, despite the fact that Asian families value strong and intimate interpersonal relationships with family members [21]. At present, Indonesia’s mental health services are not organized to deliver specific programs accordingly to each patient’s needs. Mental health services in Southeast Asia have often been reported to lack: (1) sufficient inpatient care, (2) case managers, (3) family-centered care, (4) skilled manpower, and (5) one-to-one time spent with patients, all factors crucial for recovery [22,23]. Therefore, it is highly recommended for policymakers in Indonesia to develop innovative programs applicable to the Indonesian population. 

There were no dropouts in this study. This can be explained by the high interest of the participants and the strict screening procedures we undertook. The IPQS-R tool was identified as a valid and reliable measure for assessing cognitive representations of mental health problems. All the subscales used showed high acceptability levels for internal consistency and stability over time and were largely independent of each other. The anonymity of this study (due to stigma) prevented interaction among family members, which was originally planned. There were more male than female patients in this study, but that is not unusual in schizophrenia studies [24]. This study had a relatively small number of participants (64 family members, 64 patients), though that was sufficient for PLS-PM to build a cause-and-effect model. The strength of this study lies in its experimental design, which includes: randomness clinical trials, double-blind conditions during data collection, and standardized guidelines and protocols. 

## 5. Conclusions

This study supports its initial hypothesis, that a 12-week course of psychoeducation provided to family members of persons with schizophrenia improves their knowledge, lowers their expressed emotion, and improves the patients’ adherence to their medical regimen. This has important implications for the provision of mental healthcare in Indonesia. 

## Figures and Tables

**Figure 1 ijerph-18-07522-f001:**
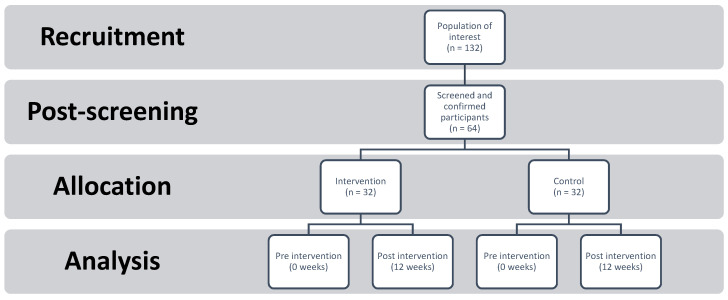
Randomized controlled trial (RCT) methodology’s adaption for this study.

**Figure 2 ijerph-18-07522-f002:**
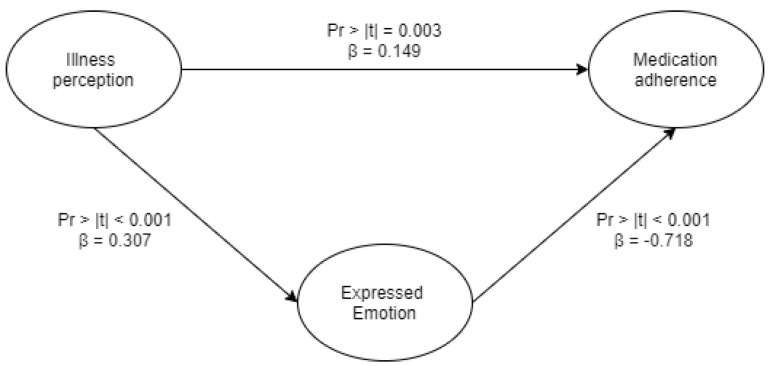
PLS-PM explaining the overall relationship between illness perception, expressed emotion, and medication adherence. Cronbach’s alpha, Dillon-Goldstein’s rho, and the R^2^ values for each variable were greater than 0.7, with a GoF of 0.739 indicating that the model is reliable.

**Table 1 ijerph-18-07522-t001:** Mean scores, control, intervention, pre-post scores, significance of effect, inter-item correlation, Cronbach’s alphas, and Kaiser-Meyer-Olkin (KMO) measures of sampling adequacy for each subscale of the IPQS-R.

IPQS-R Subscales	Control		Intervention		Significance of Effect ^b^	Effect Size (Control; Intervention) ^c^	Mean Inter-Item Correlation	Cronbach Alpha	KMO
	Pre	Post	Pre	Post					
Identity ^a^	9.04 (1.93)	9 (1.71)	8.97 (1.86)	7.1 (2.5)	F_(1,62)_ = 5.85 *	d = 0.02; 0.85	-	-	-
Cause Items	2.91 (0.28)	3 (0.24)	2.99 (0.31)	1.5 (0.16)	F_(1,62)_ = 696.18 ***	d = 0.35; 6.04	0.25	0.88	0.76
Timeline Acute	2.67 (0.88)	2.7 (0.9)	3.75 (1.09)	2.87 (1.08)	F_(1,62)_ = 9.8 ***	d = 0.03; 0.81	0.45	0.93	0.64
Timeline Cyclical	4.11 (0.73)	4.13 (0.68)	4.24 (0.58)	4.07 (0.62)	F_(1,62)_ = 4.65 *	d = 0.03; 0.28	0.7	0.83	0.56
Consequence—Patient	3.88 (0.5)	3.98 (0.48)	3.9 (0.48)	3.74 (0.43)	F_(1,62)_ = 7.07 **	d = 0.2; 0.35	0.23	0.57	0.66
Consequence—Relatives	3.58 (0.32)	3.56 (0.3)	3.51 (0.39)	1.76 (0.2)	F_(1,62)_ = 216.17 ***	d = 0.06; 5.65	0.33	0.91	0.73
Personal Control—Patient	1.7 (0.43)	1.72 (0.47)	1.9 (0.43)	3.28 (0.56)	F_(1,62)_ = 76.45 ***	d = 0.04; 2.76	0.38	0.9	0.55
Personal Control—Relatives	1.77 (0.52)	1.53 (0.42)	1.75 (0.55)	2.56 (0.85)	F_(1,62)_ = 25.86 ***	d = 0.51; 1.13	0.36	0.82	0.6
Personal Blame—Patient	3.88 (0.6)	3.79 (0.69)	3.94 (0.89)	2.37 (0.54)	F_(1,62)_ = 25.4 ***	d = 0.14; 2.13	0.31	0.86	0.64
Personal Blame—Relatives	2.02 (0.35)	2.1 (0.35)	2.02 (0.3)	3.03 (0.45)	F_(1,62)_ = 72.17 ***	d = 0.23; 2.64	0.36	0.82	0.62
Treatment Control	2.43 (0.43)	2.61 (0.34)	2.53 (0.45)	3.53 (0.68)	F_(1,62)_ = 20.46 ***	d = 0.46; 1.73	0.25	0.85	0.54
Illness Coherence	3.82 (0.46)	3.81 (0.53)	2.69 (0.31)	3.84 (0.45)	F_(1,62)_ = 51.1 ***	d = 0.02; 2.98	0.39	0.91	0.72
Emotional Representation	3.64 (0.28)	3.69 (0.29)	3.71 (0.29)	1.86 (0.15)	F_(1,62)_ = 411.96 ***	d = 0.18; 8.01	0.37	0.96	0.59

^a^ The cumulative score of Yes or No responses. ^b^ Values of F-scores from RM-ANOVA for pre-post control and intervention groups. * Indicates significance at 5% level, ** 1% level, and *** 0.1% level. ^c^ Calculated effect size based on d-value of difference between group and standard deviation.

**Table 2 ijerph-18-07522-t002:** Number of family members with high expressed emotion, as assessed by a five-minute speech sample (FMSS).

Family Members’ FMSS Expressed Emotion	Control		Intervention		Odds Ratio
	Pre	Post	Pre	Post	
Criticism	14	15	20	7	0.32 (CI: 0.11–1.01)
Emotional overinvolvement	19	17	12	5	0.46 (CI: 0.14–1.6)

Note: Family members can be both rated as high in criticism and emotional overinvolvement; therefore, the number of family members in the control group pre-intervention may add up to more than 32 participants for each group. The table only lists the number of family members that expressed a high level of expressed emotion; therefore, the number of family members in the post-intervention group may not add up to 32 participants.

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
