# Peer review of "Psychoeducation Improved Illness Perception and Expressed Emotion of Family Caregivers of Patients with Schizophrenia"

_ijerph, 2021, doi:10.3390/ijerph18147522_

Round 1

Reviewer 1 Report

The manuscript has been much improved. The discussion is still a bit verbose, which I don't like very much, but I don't judge it unworthy of publication because of that.

However, some minor corrections are still needed, as shown below.

  1. Both the terms "caretaker" and "caregiver" were used. Was this intentional?
  2. In this revision, the authors have added to the previous literature on psychoeducation (line 78 to 87), which included many studies that provided psychoeducation to the patients themselves. Since this study intervened only with families of patients, the review should focus on studies that intervened with families.
  3. The sentence "The success of family intervention can be attributed to the benefit of increased compliance and adherence from patients with appropriate social and cognitive context [21] where both are reported to influence the success of a treatment [22]." written by the authors in the responses to reviewer could not be found in the text. I think it was accidentally deleted, so please correct it.
  4. The authors described that "RCTs had reported lowered relapse rate and readmission at all time points [9-11]" (line 81 to 82). What were "all time points?" I would recommend simply removing this.
  5. There are some places where expressed emotion is written as EE and some places where it is not. I think it would be easier for readers to understand if the authors did not abbreviate this.
  6. In addition, there were a few places where it was unclear whose expressed emotion it was (line 382, 414, 415 and 475).
  7. The authors described that "Another study had also showed that sessions with both ..... was significantly lower compared to the control group with 75% relapse rate [63]" (line 423 to 427). These sentences are confusing and should be summarized by focusing only on the effects of psychoeducation.
  8. Strength and limitation should be written at the end of the discussion, not in the conclusion.

Author Response

Please find responses attached

Reviewer 2 Report

I have no further comments or revisions to the changes already made by the authors.

Author Response

We'd like to thank the reviewer for their comments

Round 2

Reviewer 1 Report

The authors have satisfactorily addressed all the reviewer concern.

Author Response

We'd like to thank the reviwers for their comment

This manuscript is a resubmission of an earlier submission. The following is a list of the peer review reports and author responses from that submission.

Round 1

Reviewer 1 Report

This study presents the effectiveness of an online psychoeducational session to families of patients with schizophrenia, which improved not only the illness perception and expressed emotion of their families, but also the patients' adherence to treatment. The results themselves are very interesting; however, there are some serious concerns about the methodology. Also, the descriptions of current issues in the introduction and discussion were not sufficient. Unfortunately, I cannot recommend this article for publication at this stage.

Comments:

INTRODUCTION

  1. The differences and advantages of this study compared to previous reports were not clear.

  1. The authors stated that “Secondly, antipsychotic medication has been questioned for its effectiveness”. While it is true that there are limitations to antipsychotic treatment and that some can become resistant to treatment, this wording is misleading given that the effectiveness of antipsychotics has been well established in general.

MATERIALS AND METHODS

  1. This is the most critical point: the study design was not written in detail. First of all, the authors did not state when the study was conducted (since the diagnosis was made using the DSM-IV, I assume that the study was conducted a long time ago). There are also some major concerns. 1) No registration as an interventional study? 2) How did the authors randomized? 3) What did the authors do for the control group? Or kept them on waiting list? 4) No dropouts in the 64 subjects? Was it true?

  1. Three outcomes (IPQS-R, FMSS and IMAS) were used for the study, but there were no references for them.

  1. I could not figure out which variables were used in the path analysis: baseline results or results in the control group were included or not?

  1. “Intervention procedure” should be described just after “study design”

RESULTS

  1. I could not find the IMAS score after the intervention. I think this was the main outcome, so the results should be well documented.

  1. Many of the contents of line 263 to 280 should be written in the discussion.

Discussion

  1. Overall, the discussion was not sufficiently detailed. In particular, there is little discussion on the main outcome of this study: the relation between the adherence of patients and their families’ perception or emotional expression. What makes the change in caregivers’ perception or emotional expression differentiates the adherence of patient can be a core of this paper, considering its clinical significance. This should be thoroughly described in the discussion.

Author Response

Reviewer 1

This study presents the effectiveness of an online psychoeducational session to families of patients with schizophrenia, which improved not only the illness perception and expressed emotion of their families, but also the patients' adherence to treatment. The results themselves are very interesting; however, there are some serious concerns about the methodology. Also, the descriptions of current issues in the introduction and discussion were not sufficient. Unfortunately, I cannot recommend this article for publication at this stage.

We’d like to thank the reviewer for the comment. We have added substantial information in the Introduction, Methodology, and Discussion as recommended below.

INTRODUCTION

  1. The differences and advantages of this study compared to previous reports were not clear.

This has been clarified highlighting that previous psychiatry studies that has been done in Indonesia focuses mainly in understanding the perception of beliefs and caring experience without taking into account the importance of illness perception, expressed emotion, and medication adherence.

  1. The authors stated that “Secondly, antipsychotic medication has been questioned for its effectiveness”. While it is true that there are limitations to antipsychotic treatment and that some can become resistant to treatment, this wording is misleading given that the effectiveness of antipsychotics has been well established in general.

We agree with the reviewer that this is rather misleading, and the authors have decided to delete the sentence and focus in introduction family intervention, tools, and its relationship in this study.

MATERIALS AND METHODS

  1. This is the most critical point: the study design was not written in detail. First of all, the authors did not state when the study was conducted (since the diagnosis was made using the DSM-IV, I assume that the study was conducted a long time ago). There are also some major concerns. 1) No registration as an interventional study? 2) How did the authors randomized? 3) What did the authors do for the control group? Or kept them on waiting list? 4) No dropouts in the 64 subjects? Was it true?

We have noticed that DSM-5 was actually used in this study, not DSM-IV and ICD-10. This has been changed accordingly in the manuscript. The data collection for this study was collected in mid-2016 the information has been added to Section 2.1.

Registration was done in Indonesia’s local health board authority. The authors were previously unaware that an additional registration is required with NIH.

Each participant who have signed up were assigned a three digit random code and was then randomised into the two groups this information blinded to the researchers has been added to Section 2.2.

The control group did not receive any psychoeducation and with one of the co-author (WB) checking in with the participants at 0 and 12 weeks – they were indeed placed in a ‘waiting list’. The participants in control group was then provided the intervention and materials after the 12 weeks period (if they wish to). This information has been added in Section 2.2.

There were a lot of interest during the study advertisement. We were lucky that no participants dropped out in this study – however, there was a dropout throughout the recruitment and screening process. This has been added in Figure 1. We have also add a commentary on this in Section 4.5.

  1. Three outcomes (IPQS-R, FMSS and IMAS) were used for the study, but there were no references for them.

This has been added accordingly in Section 2.2 with substantial information on the dimensions and sub-scales used.

  1. I could not figure out which variables were used in the path analysis: baseline results or results in the control group were included or not?

Both pre-post control and intervention groups was included in the path analysis model. This detail has been added in Section 2.5.

  1. “Intervention procedure” should be described just after “study design”

This has been changed as recommended and now parked under Section 2.2. Additionally, more information on the psychoeducation has been added to this section.

RESULTS

  1. I could not find the IMAS score after the intervention. I think this was the main outcome, so the results should be well documented.

The results from IMAS can be found descriptively with now mean values and F values added in Section 3.5. Following with the other reviewer’s comment the analysis in this study has been reworked and more details has been added to the manuscript.

  1. Many of the contents of line 263 to 280 should be written in the discussion.

We agree and have relocated this in Discussion section.

DISCUSSION

  1. Overall, the discussion was not sufficiently detailed. In particular, there is little discussion on the main outcome of this study: the relation between the adherence of patients and their families’ perception or emotional expression. What makes the change in caregivers’ perception or emotional expression differentiates the adherence of patient can be a core of this paper, considering its clinical significance. This should be thoroughly described in the discussion.

We agree with this comment although a brief discussion for the translated tool in this study is required (Section 4.1)

We have now rewritten the Discussion section and have sub-headed the section to highlight how the change in perception and expressed emotion in the caregivers resulted in increased medication adherence with supporting references.

Reviewer 2 Report

This paper addresses the topic of treatment with psychoeducation in order to increase adherence of people with schizophrenia to the medication programme. The study is correct, with randomised between-groups design, and the theme and development areappropriate,but they are some problems in the text that should be improved, especially the rationaleand the data analyses. The article could be published with major changes. Content aspects: Shorter title and more concrete. The title is confusingbecause we do not know if the improvement is about schizophrenia people, their families, or the medication adherence. Perhaps “Results ofpsychoeducational programmeabout relatives of peoplewith schizophrenia”.

Better abstract. It is too general, it should describe information about participants, groups, measurements, quantitative results and significance (e.g.,effect size).

The introduction deals with treatment of schizophrenia problems and psychological treatment, but anything about the fundamental goals of the study, i.e., perceptions and emotions from families, and medication adherence from people with those problems. The review of literature should be reconsidered.

Also, the relationship between family attitudes and medication adherence is not justified. Why should it improve?This relationshipshould be reconsidered and justified in that introduction.

The participants (relatives) were not really randomised, they were volunteers interestedin the programme. Really,they were families, not patients. Also,they were recruited from social networks. This doesn’t appear in abstract.

It is not clear whether patients were also interviewed, or only relatives?

The authors do not mention drop-out from the study, itis rare. When it is common that not as many participants answer at the end as at the beginning.

The instruments should be described better (how much items, subscales, reliability, validity)

The design should be explainedbetter. It is a factorial 2x2 design, and the data analysis should be in correspondence with this design.

The data should be compared with a Factorial Analysis, comparing between control and experimental group, with pre-post data as intra-group. The authors had made a pre-post t-Student analysis, and it is common that the results were significant, but the real differences must be between both groups, not pre-post. So, the data analysis should be re-done, and for all measurements.

It is not relevant the reliability and correlation between subscalesoritems, because the sample is only 32 participants.

The description of the intervention procedure is too short. That is the part more interesting for potential readers: What exactly they did in this educationalprogramme?

In Table 1 it appears in second column the N. Is it the participants or the items number?

In Table 2 it appears N=128. What? Participants? Items?If the latter, it is totally incorrect. The N always refers to participants or individuals completing a questionnaire.

Why use a Chi2 to compare EE? It should be between control and intervention group, with ODDS probability, not only pre-post

The IMAS is an adaptation of Morisky MAS, so why the authors score the items inversely?It is assumed that the higher the adherence, the higher the value.

Again, the comparison with IMAS should be between control and intervention group, not pre-post.

In a similar way, a mathematical model, or latent model, or structural equations, from the questionnaires with only 32 participants is insufficient. The results presented by authors here are very risky.

The authors do no mention whether they have mixed the score of both groups, for thosePLS-PM analyses. Or they use only intervention group?

The studies described in row 260 to 315, really are the review that should be in the introduction.

The authors talkabout “double blind condition during data collection”, butthis information doesn’t appear in method.Formal aspects:

Keywords should include “schizophrenia”.

They are different font size within the text.

References into the text should follow the APA 7th edition handbook.

The description of patients and family members characteristics should be in “participants” description, not in results.

References at the end should includeDOI

Author Response

Reviewer 2

This paper addresses the topic of treatment with psychoeducation in order to increase adherence of people with schizophrenia to the medication programme. The study is correct, with randomised between-groups design, and the theme and development are appropriate, but they are some problems in the text that should be improved, especially the rationale and the data analyses. The article could be published with major changes. Content aspects: •Shorter title and more concrete. The title is confusing because we do not know if the improvement is about schizophrenia people, their families, or the medication adherence. Perhaps “Results of psychoeducational programme about relatives of people with schizophrenia”.

We’d like to thank the reviewer for their input. Resonating with the other reviewer’s comment we have rewritten the Introduction and Methodology to provide more clarity to the reads.

We agree with the title recommendation and have revised it to: “Psychoeducation improved illness perception and expressed emotion of family caregivers of patients with schizophrenia”

Better abstract. It is too general, it should describe information about participants, groups, measurements, quantitative results and significance (e.g., effect size).

More details on the results of this study alongside with path analysis results has been incorporated in the abstract as recommended.

The introduction deals with treatment of schizophrenia problems and psychological treatment, but anything about the fundamental goals of the study, i.e., perceptions and emotions from families, and medication adherence from people with those problems. The review of literature should be reconsidered.

The introduction has been rewritten incorporating the concept of illness perception and expressed emotion and how it is linked to medication adherence. We have also expanded the current psychiatry studies in Indonesia to identify the gap of knowledge to highlight the underlying reason why this study is conducted.

Also, the relationship between family attitudes and medication adherence is not justified. Why should it improve? This relationship should be reconsidered and justified in that introduction.

This is now incorporated in the Introduction section highlighting how family’s attitude and behaviour is important and the link to medication adherence is explained.

The participants (relatives) were not really randomised, they were volunteers interested in the programme. Really, they were families, not patients. Also, they were recruited from social networks. This doesn’t appear in abstract.

The participants in this study were indeed the relatives of the patient where the researchers have recruited through either online media or in-person during seminars or visits. They were then assigned a three-digit code to anonymise them and then were allocated randomly in either control or intervention group. This detail has been added in Section 2.2.

The abstract now also have the information on participants recruitment.  

It is not clear whether patients were also interviewed, or only relatives?

It is only the relatives that were interviewed for the study using IPQS-R developed by Lobban et al (2005)

The authors do not mention drop-out from the study, it is rare. When it is common that not as many participants answer at the end as at the beginning.

We agree that this is indeed a rare phenomenon, we have added a substantial explanation under Section 4.5 which explains that perhaps this is due to the high interest, strict screening procedure, and multiple reminders provided by the researchers for the participants.

The instruments should be described better (how much items, subscales, reliability, validity)

More details for subscales and items can now be found under Section 2.3 for IPQS-R and IMAS.

The design should be explained better. It is a factorial 2x2 design, and the data analysis should be in correspondence with this design. The data should be compared with a Factorial Analysis, comparing between control and experimental group, with pre-post data as intra-group. The authors had made a pre-post t-Student analysis, and it is common that the results were significant, but the real differences must be between both groups, not pre-post. So, the data analysis should be re-done, and for all measurements.

We agree with the reviewer and we have run a Repeated Measures ANOVA based on this 2x2 design this detail can be found on Section 2.5 and the results section has been rewritten to illustrate these changes.

It is not relevant the reliability and correlation between subscales or items, because the sample is only 32 participants.

There is a total of 64 participants in this study and although we acknowledge that this is a rather small sample size this information would be important for the readers to show that the translation of IPQS-R to Indonesian language is sufficiently robust.

The description of the intervention procedure is too short. That is the part more interesting for potential readers: What exactly they did in this educational programme?

More information has now been added under Section 2.2.

In Table 1 it appears in second column the N. Is it the participants or the items number?

Items number for the subscale but as the reviewer has recommended this information has been integrated accordingly in Section 2.3

In Table 2 it appears N=128. What? Participants? Items? If the latter, it is totally incorrect. The N always refers to participants or individuals completing a questionnaire.

N number here refers to the total number of observations in this study which was a total of 128 observations from 64 participants.

Why use a Chi2 to compare EE? It should be between control and intervention group, with ODDS probability, not only pre-post

We agree that chi-square may not be the best approach, as recommended the odds ratio and its Cis has been calculated and integrated in the manuscript in Section 3.4.

The IMAS is an adaptation of Morisky MAS, so why the authors score the items inversely? It is assumed that the higher the adherence, the higher the value.

We agree with the reviewer that this causes confusion, a score reversion is done and the results has been rewritten.

Again, the comparison with IMAS should be between control and intervention group, not pre-post.

We agree with the reviewer and this has been changed accordingly

In a similar way, a mathematical model, or latent model, or structural equations, from the questionnaires with only 32 participants is insufficient. The results presented by authors here are very risky.

A total of 64 participants which totalled up to 128 observations were included in the model. We do agree that perhaps the nature of smaller sample size could be troublesome for classical path analysis. However, PLS-PM possess advantages with smaller sample size which can be attributed to: (1) ignoring the effects of chance correlations among measurement errors, which inflates parameter estimates; and (2) the use of a t distribution for NHST when the coefficient distributions are not normal, which leads to increased Type I error rates. As such compared to SEM and traditional analysis, PLSPM were able to produce bimodal parameter estimates with modes that were positive and negatives where SEM and summed scale approach yielded estimates at zero even at samples sizes as small as 25.

We have however added a statement for future studies to include more participants in our limitation section.

The authors do no mention whether they have mixed the score of both groups, for those PLS-PM analyses. Or they use only intervention group?

It was the scores for all groups to ensure correct representation of the results.

The studies described in row 260 to 315, really are the review that should be in the introduction.

As recommended by the previous reviewer this section has been parked in Discussion and has been rewritten.

The authors talk about “double blind condition during data collection”, but this information doesn’t appear in method. Formal aspects:

More details on how the participants and the researchers were coded now has been included in Section 2.2

Keywords should include “schizophrenia”.

This has been added accordingly

They are different font size within the text.

This has been fixed accordingly

References into the text should follow the APA 7th edition handbook.

The referencing was done by EndNote and now has been updated to the journal’s requirement following ACS guideline.

The description of patients and family members characteristics should be in “participants” description, not in results.

We disagree with the reviewer for this comment as the methodology section should retain only methods.

References at the end should include DOI

The referencing follows the format that MDPI’s IJERPH which was available downloadable as an EndNote style the referencing program that we use for this manuscript – the DOI information however has been added in the travelling library but doesn’t show up with the final manuscript.

Round 2

Reviewer 1 Report

I appreciate the authors’ efforts in revising the manuscript. The manuscript has been much improved. Yet, it still requires major revision before ready for publication.

  1. This revision by the authors increased the volume of the discussion, but at the same time made it very redundant. It is very difficult to understand the excellence of this study in this Discussion. I think the findings supporting the hypothesis of this study were the most impressive part, so this should be emphasized. For the other parts, it would be more reader-friendly to summarize. Please rewrite it extensively.

  1. In the last paragraph of the introduction, the authors seem to have focused on the Indonesian studies investigating the family psychoeducation. Since the IJERPH is an international journal, the authors should state what is already known and what is not, in light of the current state of the world as a whole.

  1. By whom was the diagnosis of schizophrenia made? By board certified psychiatrists?

  1. I understood that data of both pre-post control and intervention groups were included in the path analysis model. It would be helpful for the reader to know how many data were used in the path analysis.

Author Response

I appreciate the authors’ efforts in revising the manuscript. The manuscript has been much improved. Yet, it still requires major revision before ready for publication.

  1. This revision by the authors increased the volume of the discussion, but at the same time made it very redundant. It is very difficult to understand the excellence of this study in this Discussion. I think the findings supporting the hypothesis of this study were the most impressive part, so this should be emphasized. For the other parts, it would be more reader-friendly to summarize. Please rewrite it extensively.

We agree with the reviewer – our Discussion section had attempted to explain our results in this study with a references supporting our hypothesis in the end of each subsection with each subsection corresponding to the results of this study. We agree that the discussion subsection on EE is lacking and have expanded the paragraph. It now reads

Previous studies had already identified expressed emotion as an important factor in patients’ medication adherence [61]. In addition, high expressed emotion has been identified as a strong predictor of rehospitalization rate in patients with schizophrenia related to non-adherence towards medication [7]. A RCT which included high EE caregivers of patients with schizophrenia had found that psychoeducation intervention was effective in positively changing 75% of the experimental group families which reported a relapse rate of 8% over 9 months compared to the control with a relapse rate of 50% [62]. Another study had also showed that sessions with both including patients and excluding patients reported 33% and 36% relapse rate at two years, respectively. Patients’ caregivers that were assigned to the intervention group were reported a relapse rate of 40% which was significantly lower compared to the control group with 75% relapse rate [63].

We also have further summarised Section 4.1 and especially 4.5

  1. In the last paragraph of the introduction, the authors seem to have focused on the Indonesian studies investigating the family psychoeducation. Since the IJERPH is an international journal, the authors should state what is already known and what is not, in light of the current state of the world as a whole.

We agree with the reviewer, we have added a couple of sentences on the recent family psychoeducation in recent published literatures before the paragraph that discusses Indonesian studies. The paragraph reads

Antipsychotic medication has always been the central attention for schizophrenia treatment. However, the attention is now shifting with the inclusion of family psychoeducation intervention as it has shown to provide a positive impact on patients’ recovery. This was demonstrated on how psychoeducation has been reported as an important treatment in parallel with regular antipsychotic medication over the past decade. Patients and caregivers that was provided with psychoeducation interventions tends to have positive outcomes which includes: 1) increased level of knowledge - a  significantly higher score of level of knowledge was reported with a large effect after psychoeducation intervention with no significant differences after a 3-month follow-up [8]; 2) number of relapses - RCTs had reported lowered relapse rate and readmission at all time points [9-11]. In addition, a study [8] had showed a lower relapse rate (measured by number and dosage of antipsychotics) in the intervention group at all time points compared to the higher relapse rate of the control group; 3) reduced symptoms severity - recent studies had shown that psychoeducation had found a positive effect overtime [12], and up to 12 months [13]. Interestingly, a longitudinal study done for 7 and 10 years reported no significant difference between control and intervention groups [14, 15]; 4) internalized stigma - patients that has been diagnosed with schizophrenia experience were reported with higher feeling of stigmatization and discrimination compared to other patients with mental illness [16]. Psychoeducation has been reported in reducing feeling of stigma [17] and therefore improve their quality of life [12]; and finally 5) medication compliance [18, 19], supplemented with improvement in their social functioning [20]. The success of family intervention can be attributed to the benefit of increased compliance and adherence from patients with appropriate social and cognitive context [21] where both are reported to influence the success of a treatment [22].

  1. By whom was the diagnosis of schizophrenia made? By board certified psychiatrists?

The diagnosis of schizophrenia was made by trained and certified psychiatrists in Indonesia prior to this study who had utilised DSM-5 for their diagnosis. We have added this information on Section 2.2 which reads

“It is important to note that the patients were diagnosed by trained certified psychiatrists in Indonesia”

  1. I understood that data of both pre-post control and intervention groups were included in the path analysis model. It would be helpful for the reader to know how many data were used in the path analysis.

We agree with the reviewer, as such all data were included in the model – which includes pre-post control and intervention observations totalling to 128 observations. We have included the following sentence under Section 2.5 which reads

“In this study, both control and intervention, pre-post group data were modelled totalling to 128 observations included”

Reviewer 2 Report

The suggestion are changed, so all OK.

Author Response

We'd like to thank the reviewer for their review.